# Allergic Reactions and Cross-Reactivity Potential with Beta-Lactamase Inhibitors

**DOI:** 10.3390/pharmacy7030077

**Published:** 2019-06-28

**Authors:** Kayla R. Stover, Katie E. Barber, Jamie L. Wagner

**Affiliations:** Department of Pharmacy Practice, University of Mississippi School of Pharmacy, Jackson, MS 39216, USA

**Keywords:** beta-lactam allergy, clavulanate, sulbactam, tazobactam, avibactam, relebactam, vaborbactam, cross reactivity, hypersensitivity

## Abstract

Although beta-lactam allergies are an emerging focus of stewardship programs and interventions, less is publicly released regarding allergies to beta-lactamase inhibitors. This review presents and evaluates the data regarding allergic reactions with beta-lactamase inhibitors. Clavulanate, sulbactam, and tazobactam are beta-lactam-based beta-lactamase inhibitors that are combined with several penicillins or cephalosporins in order to preserve antimicrobial activity in the presence of beta-lactamases. Avibactam, relebactam, and vaborbactam are non-beta-lactam beta-lactamase inhibitors that are combined with cephalosporins or carbapenems in order to expand the antimicrobial activity against broader-spectrum beta-lactamases. Case reports document hypersensitivity reactions to clavulanate, sulbactam, and tazobactam, but not to avibactam, relebactam, or vaborbactam. Based on these reports and considering the chemical structures, cross-allergenicity with beta-lactams is likely with sulbactam and tazobactam. Considering the slightly altered beta-lactam structure, cross-allergenicity is less likely with clavulanate, but still possible. It appears that cross-allergenicity between beta-lactam antimicrobials and the newer, non-beta-lactam beta-lactamase inhibitors is unlikely. It is important for clinicians to perform allergy testing to both the beta-lactam and the beta-lactamase inhibitor in order to confirm the specific allergy and reaction type.

## 1. Introduction

Antimicrobial allergies are an emerging focus of many healthcare practitioners, as the presence of an allergy may result in suboptimal treatment options, increased complications related to care, and worse patient outcomes [1,2,3]. Currently, information related to beta-lactam allergies is well publicized, and this is an increasing focus of interventions and stewardship efforts [4,5,6,7].

Beta-lactamase inhibitors were first developed in the 1970s in response to the increasing beta-lactamase-mediated resistance to beta-lactam antibiotics [8]. Structurally related to penicillin (Figure 1), these inhibitors irreversibly bind to beta-lactamases, causing chemical reactions at the enzyme active site, and permanently inactivate the enzyme [8,9]. When used concomitantly, beta-lactamase inhibitors enhance the activity of beta-lactams, thereby allowing the beta-lactam to reach the target site [8]. Consequently, this expands the activity of these agents in the presence of beta-lactamases, specifically those that are class 2a [8].

Although beta-lactam allergies are a large focus currently, less is publicly known or released regarding allergies to beta-lactamase inhibitors. The purpose of this review is to present and evaluate the data regarding allergic reactions with beta-lactamase inhibitors.

## 2. Beta-Lactamase Inhibitors

### 2.1. Clavulanate

Clavulanate, or clavulanic acid, is a beta-lactam-based beta-lactamase inhibitor. The structure is similar to a penicillin nucleus, but with an oxygen substitution for sulfur to make a fused, bicyclic beta-lactam and oxazolidine ring base [10]. Possibly because of this substitution, clavulanate has a higher potency (when compared gram for gram) than some of the other beta-lactamase inhibitors [8]. Clavulanate has weak inherent antimicrobial activity, possibly due to interactions with penicillin binding proteins and the host immune system [11]. Specifically, clavulanate is known to inactivate penicillin binding proteins in *Haemophilus influenzae* and *Neisseria gonorrhoeae* [8]. Although weak, the antimicrobial spectrum of clavulanate includes activity against Gram negative and positive bacteria and anaerobes, specifically against *Bacteroides* spp., *Moraxella catarrhalis*, staphylococci, streptococci, *Neisseria* spp., *Chlamydia* spp., and *Legionella* spp. [11]. Because the antimicrobial activity is weak, clavulanate is always given in combination.

Clavulanate is commercially combined with amoxicillin in several formulations including tablets, chewable tablets, and suspensions. When combined, clavulanate increases the activity of amoxicillin against *Streptococcus pneumoniae*, *H. influenzae*, *M. catarrhalis*, *Legionella pneumophila*, *Chlamydia trachomatis*, penicillin-susceptible *Staphylococcus aureus*, *Actinobacillus actinomycetemcomitans*, *Enterococcus faecalis, Bacteroides* spp., *Escherichia coli, Klebsiella* spp., and *Proteus mirabilis* [8,11]. Amoxicillin–clavulanate is indicated for the treatment of lower respiratory tract infections, acute bacterial otitis media, and sinusitis caused by beta-lactamase-producing *H. influenzae* and *M. catarrhalis*; skin and skin structure infections caused by beta-lactamase-producing *S. aureus*, *E. coli*, and *Klebsiella* spp.; and urinary tract infections caused by beta-lactamase-producing *E. coli*, *Klebsiella* spp., and *Enterobacter* spp. [12].

### 2.2. Sulbactam

Sulbactam is also a beta-lactam-based beta-lactamase inhibitor with a penicillanic acid sulfone structure [13]. Sulbactam is less potent against class A beta-lactamases than clavulanate, but is more potent against class C beta-lactamases [13]. Although not available as a single-formulated agent, sulbactam has intrinsic activity against *Acinetobacter* spp.; *Bacteroides fragilis* [13,14]. This is likely due to its ability to bind to the penicillin-binding protein 2 of these organisms [14]. As a result, ampicillin–sulbactam is often used in the treatment of infections caused by *Acinetobacter* spp., despite ampicillin not having discernible activity [15].

Sulbactam is commercially combined with ampicillin and is available as a powder for solution for injection. When combined, sulbactam increases the activity of ampicillin against beta-lactamase-containing *S. aureus*, *H. influenzae*, *M. catarrhalis*, *E. coli, P. mirabilis*, *Salmonella* spp., and *Shigella* spp. [8,10]. Ampicillin–sulbactam is indicated for the treatment of skin and skin structure infections caused by beta-lactamase-producing *S. aureus*, *E. coli*, *Klebsiella* spp., *P. mirabilis*, *Bacteroides fragilis*, *Enterobacter* spp., and *Acinetobacter baumannii–calcoaceticus* complex; intra-abdominal infections caused by beta-lactamase-producing *E. coli*, *Klebsiella* spp., *Bacteroides* spp., and *Enterobacter* spp.; and gynecological infections caused by beta-lactamase-producing *E. coli* and *Bacteroides* spp. [16].

### 2.3. Tazobactam

Tazobactam is a beta-lactam-based beta-lactamase inhibitor with a structure similar to sulbactam, but potency more similar to clavulanate [9,17]. The antimicrobial spectrum of tazobactam in vitro includes *H. influenzae*, *M. catarrhalis*, and *Acinetobacter anitratus* [17].

Tazobactam is commercially combined with piperacillin and ceftolozane and is available as a powder for solution and injection. When combined, tazobactam increases the activity of piperacillin against Enterobacteriaceae, *H. influenzae*, *N. gonorrheae*, and *M. catarrhalis*, and may lower the minimum inhibitory concentration (MIC) against organisms producing extended-spectrum beta-lactamases [8]. Piperacillin–tazobactam is indicated for the treatment of intra-abdominal infections, skin and skin structure infections, female pelvic infections, community-acquired pneumonia, and nosocomial pneumonia [16]. When combined with ceftolozane, activity includes *Enterobacter cloacae*, *E. coli*, *Klebsiella* spp., *P. mirabilis*, *Pseudomonas aeruginosa*, *B. fragilis*, and *Streptococcus* spp. [16]. Ceftolozane–tazobactam is indicated for the treatment of complicated intra-abdominal infections (in combination with metronidazole), complicated urinary tract infections including pyelonephritis, and hospital-acquired and ventilator-associated bacterial pneumonia [16].

### 2.4. Avibactam

Avibactam is a synthetic diazabicyclooctane non-beta-lactam beta-lactamase inhibitor [18]. A more potent inhibitor than the older agents, avibactam has activity against class A, C, and some class D beta-lactamases, including extended-spectrum, AmpC, KPC, and OXA-48 beta-lactamases. It is thought that this activity is due to a non-covalent binding at the beta-lactamase binding site, then a second step at the serine residue featuring a covalent acylation [18].

Avibactam is commercially available in combination with ceftazidime. When combined, ceftazidime–avibactam has increased activity against multidrug-resistant *Pseudomonas aeruginosa*, *Bacteroides fragilis*, *Clostridium perfringens*, *Prevotella* spp., and *Porphyromonas* spp. Despite this increased activity, ceftazidime–avibactam is not thought to have reliable activity against anaerobic pathogens [18]. Ceftazidime–avibactam is indicated for complicated intra-abdominal infections in combination with metronidazole, complicated urinary tract infections including pyelonephritis, and hospital-acquired and ventilator-associated bacterial pneumonia [16].

### 2.5. Relebactam

Relebactam is a bicyclic urea beta-lactamase inhibitor with a structure similar to avibactam [18]. In trials, it was shown to have activity against Ambler class A and C beta-lactamase enzymes.

Relebactam is being studied in combination with imipenem–cilastatin. When combined, imipenem–cilastatin has additional activity against KPC enzymes, including imipenem-resistant Enterobacteriaceae and *P. aeruginosa* [18]. Although not yet approved by the Food and Drug Administration (FDA), imipenem–relebactam received designation as a Qualified Infectious Diseases Product with Fast Track status. Ongoing and completed phase III studies for imipenem–cilastatin–relebactam include complicated intra-abdominal, complicated urinary tract, and hospital-acquired and ventilator-associated bacterial pneumonia [19].

### 2.6. Vaborbactam

Vaborbactam is a boronic acid beta-lactamase inhibitor with activity against class A and C enzymes, including KPC beta-lactamases [18]. As a boronic acid inhibitor, vaborbactam binds the serine and boronate moiety covalently, acting as a competitive beta-lactamase inhibitor. 

Vaborbactam is commercially available in combination with meropenem. When combined, meropenem–vaborbactam has increased activity against carbapenemase-producing Enterobacteriaceae including *E. coli*, *Klebsiella pneumoniae*, and *Enterobacter* spp. However, no improvement in the base activity of meropenem was seen in either *P. aeruginosa* or *A. baumanii* [18]. Meropenem–vaborbactam is indicated for adult patients with complicated urinary tract infections including pyelonephritis [16].

## 3. Beta-Lactamase Inhibitor Allergic Reactions

### 3.1. Clavulanate

When clavulanate was first marketed as a beta-lactam-based inhibitor in combination with amoxicillin, it was thought to be non-immunogenic, and allergic reactions that occurred were attributed to amoxicillin or penicillin [20]. However, an intermediate clavulanate metabolite was shown to elicit immunoglobulin E (IgE) hypersensitivity reactions [21]. Additionally, because clavulanate is not commercially available as a single agent, allergic reports are uncommon.

In a study of 51 patients with proven immediate hypersensitivity to either amoxicillin or clavulanate, patients were assessed via skin testing, drug provocation testing, and re-provocation testing [22]. A total of 11 patients were determined to be clavulanate-selective responders and had tolerance to penicillin G, penicillin V, and amoxicillin, demonstrating that patients can have clavulanate allergies but safely take penicillin derivatives. Ten pediatric patients diagnosed with a clavulanate allergy with specific IgE testing and skin-prick testing were negative for penicillin G, penicillin V, amoxicillin, ampicillin, and cefaclor reactions [23]. All patients were able to tolerate seven days of oral amoxicillin without issue. However, receipt of amoxicillin–clavulanate led to urticaria, angioedema, or urticarial angioedema. Additionally, nine adult patients were reported to have clavulanate-selective allergic reactions [24]. These patients had previous immediate reactions to amoxicillin–clavulanate and were referred for further allergy evaluation. All nine patients had negative skin-prick and intradermal tests to benzylpenicillin, amoxicillin, ampicillin, cefuroxime, and ceftazidime. Intradermal tests to amoxicillin–clavulanate were positive in eight out of nine patients. Those eight patients then received oral challenge with amoxicillin which resulted in negative reactions. Additionally, purified clavulanate was utilized in skin-prick and intradermal tests on these patients, and seven out of nine had positive results, although the authors hypothesized that the negative results from the other two patients were likely due to a five-year time period between the original reaction and the testing.

Similarly, at another allergy department, 55 of 276 patients assessed for beta-lactam allergies via skin test had positive results [25]. These 55 patients underwent further assessments with drug provocation testing using benzylpenicillin, amoxicillin, and amoxicillin–clavulanate. Tolerance to both benzylpenicillin and amoxicillin was observed in seven patients which were then deemed allergic to clavulanate. Lastly, a patient that developed urticarial, facial angioedema and dyspnea within 15 min of amoxicillin–clavulanate receipt received an allergic workup a year post-reaction [26]. Due to negative skin-prick and intradermal tests, drug provocation tests were performed with both amoxicillin and amoxicillin–clavulanate. Amoxicillin yielded negative results; however, after a cumulative dose of 16 mg of amoxicillin–clavulanate, she developed urticaria, conjunctivitis, throat swelling, and hypotension. A 46-year-old woman developed erythematous macules and papules over her abdomen within 24 h of receiving amoxicillin–clavulanate [27]. Upon allergy testing, she had a positive skin-prick test to amoxicillin–clavulanate but not amoxicillin alone. The authors concluded that the drug rash was a T-cell-mediated eruption to clavulanate. Another incident was reported with a 27-year-old woman who received amoxicillin–clavulanate for sinusitis [28]. Within one week, the patient developed an immediate hypersensitivity to the drug. She was tested via skin prick against penicillin, which yielded negative results. When tested via the scratch method against amoxicillin–clavulanate, the reaction was clearly positive. The patient was then tested against pure clavulanate via the skin-prick method and demonstrated a positive result. Lastly, a 25-year-old patient who underwent bariatric surgery developed a delayed anaphylactic reaction to amoxicillin–clavulanate four hours after receiving a dose [29]. The patient underwent allergy testing for both amoxicillin and amoxicillin–clavulanate. An oral challenge with amoxicillin was tolerated, and skin-prick and intradermal tests with differing concentrations of amoxicillin–clavulanate were also tolerated. However, upon an oral challenge with amoxicillin–clavulanate, the patient developed an anaphylactic reaction within four hours of ingestion.

While the majority of patients demonstrate either an allergic response to amoxicillin or to clavulanate, a report of four patients described that reactions to both amoxicillin and to clavulanate can occur in the same patient [30]. In this assessment, skin tests and drug provocation tests were performed. Additionally, in the three patients with immediate reactions, specific IgE and basophil activation were analyzed. Two of these three patients had positive skin tests to both amoxicillin and to clavulanate. The third patient had a positive skin test for clavulanate and a positive drug provocation test to amoxicillin. All three patients had positive basophil activation tests to amoxicillin and to clavulanate, but negative results for benzylpenicillin.

Several testing methods are available for suspected clavulanate allergy. The most commonly used allergy testing method is the skin test. Unfortunately, in patients with a clavulanate allergy, the skin test is positive in only 9–18.7% of patients, thereby limiting its usefulness [31]. When clavulanate is tested intradermally, there is a 63.6–81.2% sensitivity to detecting a positive allergy. The basophil activation test is useful for determining an IgE-mediated reaction, especially with amoxicillin–clavulanate, with 50% sensitivity and 90% specificity. Another methodology, the drug provocation test, can be used to help detect clavulanate allergy in difficult to detect cases; however, the complexity and technical training required to conduct this test limit its use in everyday situations. The last testing method available for detecting a clavulanate allergy is the histamine release test [21]. This test has variable results due to the technical difficulties of conducting it (e.g., large blood volume, detecting histamine within the sample, and requiring leukocyte enrichment to increase the sensitivity of the test), thereby also limiting its usefulness in everyday situations. It also has less than 60% sensitivity to detecting an immediate allergy to clavulanate.

In patients receiving amoxicillin–clavulanate that experience allergic responses, these reports indicate that the reaction may be from the clavulanate component. If possible, intradermal and/or basophil activation tests should be performed in patients experiencing an allergy to amoxicillin–clavulanate to differentiate whether the response is attributed to amoxicillin versus clavulanate.

### 3.2. Sulbactam

Twelve reports were identified in which patients had reactions following administration of either ampicillin–sulbactam or cefoperazone–sulbactam [32,33,34,35,36,37,38,39,40,41,42,43]. Only one of these reports was definitely associated with the sulbactam component of the antibiotic. In this case, a nurse responsible for drug handling and preparation experienced contact urticaria from sulbactam and allergic contact dermatitis from ampicillin [32]. These reactions were confirmed by patch and scratch tests, which revealed strong positive reactions to ampicillin–sulbactam and amoxicillin–clavulanate on the patch test, an immediate reaction to sulbactam on the scratch test, and a delayed vesicular eczematous reaction to ampicillin on the scratch test.

In five reports, authors made no specific assessment on whether the allergic reaction was due to the beta-lactam or the inhibitor (or both). In the first case, the patient experienced anaphylactic reaction resulting in coronary artery spasm after administration of cefoperazone–sulbactam [33]. Pre-procedure allergy testing for cefoperazone–sulbactam was negative, but a lymphocyte transformation test completed three weeks later was highly positive. In the second case, the patient experienced Kounis syndrome after ten minutes of an ampicillin–sulbactam infusion [34]. In the third report, the patient experienced acute localized exanthematous pustulosis with multiple pustules and underlying erythema after receipt of cefoperazone–sulbactam [35]. In the fourth case, the patient experienced erythema, vesicles, and blisters with biopsy-proven linear deposits of IgA on the dermal–epidermal junction after seven days of therapy with ampicillin–sulbactam [36]. In the next report, the patient experienced respiratory distress and loss of consciousness minutes after the start of a cefoperazone–sulbactam infusion [37]. Investigators analyzed blood concentrations of the two drugs using liquid chromatography coupled with mass spectrometry, demonstrating that the death was caused by hypersensitivity and not overdose. They proposed that this may be a useful method in the future evaluation of anaphylaxis.

In the final six reports, the authors either attributed the reaction to ampicillin or cefoperazone, or there was evidence to suggest that reactions were caused by the beta-lactam (as opposed to sulbactam) [38,39,40,41,42,43]. In one case, the patient experienced acute generalized exanthematous pustulosis after receipt of amoxicillin, ampicillin–sulbactam, ciprofloxacin, clindamycin, meropenem, and vancomycin [38]. Because the patient initially received amoxicillin and ciprofloxacin with subsequent reactions to both of these, the authors concluded that this reaction was caused by one of these agents. In the second case, the patient experienced Baboon syndrome after administration of ampicillin–sulbactam [39]. Although it is possible that this was caused by either component, the authors reported that usual culpable agents include amoxicillin, ceftriaxone, and penicillin, suggesting that ampicillin was the responsible party. In the third case, the patient experienced hypersensitive vasculitis after receipt of cefoperazone–sulbactam [40]. Lesions were characterized by mononuclear cell infiltration, but disappeared within seven days of drug discontinuation. The authors attributed this reaction to the cefoperazone. In the final three cases, patients experienced myocardial injury or infarction that were suspected to be caused by anaphylactic reactions following administration of ampicillin–sulbactam [41,42]. In the first case, the patient experienced ST-segment elevation with anterior chest pain within 10 minutes of the start of the ampicillin–sulbactam infusion [41]. The authors reported that it was learned later that the patient had a previous history of urticaria and angioedema following penicillin, suggesting this was more likely caused by ampicillin than sulbactam. In the second case, the patient experienced syncope within 15 minutes of ampicillin–sulbactam ingestion [42]. The authors ordered circulating specific IgE levels for ampicillin, which were moderately positive. No specific tests were completed to assess the reaction to sulbactam specifically. In the final case, the patient experienced allergic angina syndrome (Kounis syndrome) that recurred after simultaneous use of amoxicillin–clavulanate and ampicillin–sulbactam [43]. The authors attributed this reaction to the combination of amoxicillin and ampicillin.

Based on these reports, it is possible that sulbactam may be responsible for allergies when combination beta-lactam (ampicillin or cefoperazone) and sulbactam products are implicated. In addition, it seems likely that the specific allergy may vary between the beta-lactam and sulbactam. When treating patients who experience allergies to one of these products or who have an allergy history to ampicillin–sulbactam or cefoperazone–sulbactam, clinicians should be aware that either component may be responsible. Allergy testing, by patch, scratch, intradermal, or skin-prick test, should be performed using both components of the combination agent in order to verify the allergy and response.

### 3.3. Tazobactam

A single case report recorded a patient with documented jaundice and hemolytic anemia that was later attributed to tazobactam [44]. After four days of receiving piperacillin/tazobactam, the 14-year-old patient developed hyperbilirubinemia, had an elevated lactate dehydrogenase, and no longer had a hemoglobin response following transfusion. Using the indirect antiglobulin reaction test, strong reactions were documented to tazobactam but not to piperacillin. While this reaction was documented, it was also stated that this was a non-immunological adsorption of the drug plasma proteins onto red blood cells.

### 3.4. Avibactam, Relebactam, and Vaborbactam

No reports or cases were found documenting allergic reactions to these agents at this point.

## 4. Cross-Reactivity and Testing for Beta-Lactamase Inhibitor Allergies

Clavulanate, sulbactam, and tazobactam are all beta-lactam-based beta-lactamase inhibitors. Therefore, there is potential for cross-reactivity between these beta-lactamase inhibitors and other beta-lactams. However, due to clavulanate’s lack of a side chain and oxazolidine ring bound to the beta-lactam ring, it was shown to not cross-react with other beta-lactams [31]. Therefore, patients with an allergy to clavulanate should still be able to safely receive other beta-lactams, such as amoxicillin and penicillin. In contrast, sulbactam’s structure is much more similar to the penicillin nucleus. As demonstrated in one report with specific testing [32], it is likely that patients with an allergy to sulbactam will cross-react with other beta-lactams with the similar core. Tazobactam was not reported to cause an immunologic reaction; however, specific testing of beta-lactam exposure after tazobactam reaction was not conducted. Based on the structure of the newer non-beta-lactam-based inhibitors (e.g., avibactam, relebactam, vaborbactam), it is assumed that there would be minimal risk for cross-reactivity to beta-lactams.

No reports tested or evaluated the potential for cross-reactivity between beta-lactamase inhibitors, but the potential exists for some cross-reactivity based on the chemical structure and known activity. Sulbactam and tazobactam both have a penicillanic sulfone structure. Similarly, avibactam and relebactam have structural similarities in the form of a diazabicyclooctane component. Because of the similarities, there is potential for cross-reactivity if a patient has an allergy to one of these agents. Based on unique structures, it is unlikely that clavulanate or vaborbactam will cross-react with any of the other beta-lactamase inhibitors.

Despite the beta-lactam component of the beta-lactam/beta-lactamase inhibitor combinations being the most implicated in allergic reactions, it is important to remember that the inhibitor might also be the inciting agent. Therefore, it is important to test for a reaction to not only the beta-lactam but also the inhibitor.

## Figures and Tables

**Figure 1 pharmacy-07-00077-f001:**
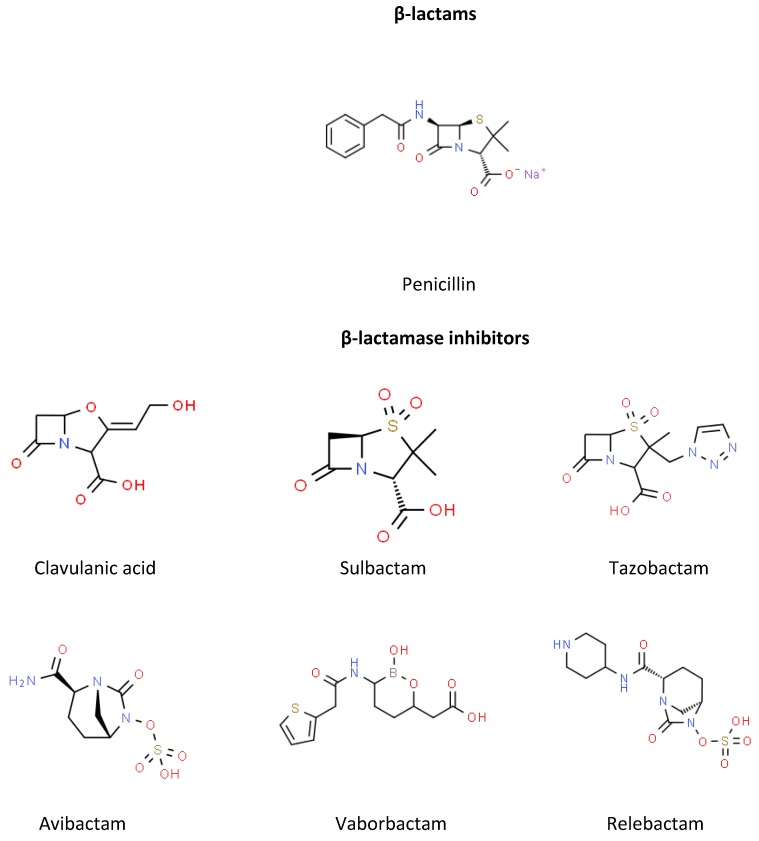
Chemical structures of the beta-lactams and beta-lactamase inhibitors.

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
