# Peer review of "Allergic Reactions and Cross-Reactivity Potential with Beta-Lactamase Inhibitors"

_pharmacy, 2019, doi:10.3390/pharmacy7030077_

Round 1
Reviewer 1 Report
In the manuscript „Allergic Reactions and Cross-Reactivity Potential with Beta-lactamase Inhibitors“ of Kayla R. Stover, Katie E. Barber, and Jamie L. Wagner, the authors reported on allergies caused by beta-lactamase inhibitors. The review summarizes and evaluates data on allergic reactions based on beta-lactamase inhibitors. The author provide data on hypersensitivity against different antimicrobial substances based on case reports. Based on its chemical structure the authors suggested that crossallergenicity with beta-lactams is likely with sulbactam and tazobactam. Information on allergenic reaction caused by antimicrobail substances are of reat importance for clinicians and pathogen treatment in humans.
This reviewer feel that the provided data are very interesting for a broad readership. However, the autors need to carefully revise the manuscript in ordert o prevent spelling and style errors.
Minor comments:
- Spelling of beta-lactam vs. Beta-Lactams (Inhibitors/inhibitors) need to be adapted to a uniform style
- Line 62: remove the dot after Escherichia
- Line 80-82: spp. not in italic lettters
- Line 83: lactamase-producing vs lactamase producing (adapt to a uniform style all over the manuscript)
- Line 114: „class A and C enzymes“ please explain or rephrase for better understanding
- Authors contribution need to be specified.
Overall, the manuscript is well written and need only minor revisions.
Reviewer 2 Report
Very well written and very informative review on the potential of beta-lactamase inhibitors to cause clinically significant allergic reactions. The authors present a lot of information in an organized fashion and help the reader to understand the extend of the current knowledge in this filed.
One way to potentially improve the value of the review for the reader is to include editorial comments in certain sections of the review. For example, both sections 3.1 and 3.2 present a lot of valuable information. An attempt to synthesize this information at the end of each section and present ideas on 1. what the next steps should be in order for the clinicians to use the current knowledge effective way and/or 2. how should we focus our research efforts to improve our knowledge in these areas, would improve, in my opinion the quality of this review.
